# Adequate vitamin A rich food consumption and associated factors among lactating mothers visiting child immunization and post-natal clinic at health institutions in Gondar Town, Northwest Ethiopia

**Addisalem Damtie Aserese**[ID][1]\*, **Azeb Atenafu**[2], **Mekonnen Sisay**[2], **Muluken Bekele Sorrie**[1], **Birhanu Wubale Yirdaw**[2], **Martha Kassahun Zegeye**[3]

1 Arbaminch University, Arba Minch, Ethiopia, 2 University of Gondar, Gondar, Ethiopia, 3 Ambo University, Ambo, Ethiopia

\* addis2010ec@gmail.com

**Data Availability Statement:** All relevant data are within the manuscript and its Supporting

## Abstract

### Background

Vitamin A deficiency is highly prevalent in low-income countries and is a major public health problem worldwide. Lactating mothers are the most vulnerable population group to vitamin A deficiency. Despite this, there is limited study on vitamin A-rich food consumption by lactating mothers in Ethiopia. Therefore, this study aimed to assess adequate vitamin A rich food consumption and associated factors among lactating mothers visiting child immunization and postnatal care centers in health institutions of Gondar Town.

### Methods

An Institution-based cross-sectional study design was employed at a health institution in Gondar Town from February to March 2019, and included 631 study participants. Simple random sampling followed by a systematic sampling technique was used to select participants. The data were collected using the Helen Keller International Food Frequency Questionnaire, entered using Epi-Info 7 statistical software and exported to STATA version 14 for analysis. A multivariable logistic regression analysis was used to identify factors associated with the outcome variable and variables with p-value <0.05 were considered as statistically significant.

### Result

A total of 624 lactating mothers participated in the study giving a response rate of 98.89%. The study shows adequate consumption of vitamin A-rich food was 38.94% (95% CI: 35%-43%). Predictors such as attending college diploma and above (AOR = 2.26, 95% CI; 1.02–4.99), having household family size $\leq 3$ (AOR = 4.04, 95% CI; 1.60–10.17), being in higher economic class (AOR = 1.93, 95% CI; 1.18–3.14), having dietary diversity score of $\geq 5$

Information files. The file name is "Vitamin A data. dta".

**Funding:** The author(s) received no specific funding for this work.

**Competing interests:** The authors have declared that no competing interests exist.

**Abbreviations:** AHC, Azezo Health Center; ANC, Antenatal care; AOR, Adjusted odds ratio; COR, Crude odds ratio; DGLVs, Dark green leafy vegetables; EPI, Expanded immunization program; EPI-INFO, Epidemiological Information; FAO, Food and Agriculture Organization; FFQ, Food frequency questionnaire; GUCSRH, Gondar University comprehensive specialized referral hospital; IQR, Interquartile range; MHC, Maraki Health Center; NGO, Non-governmental organization; PHC, Poly Health Center; PNC, Postnatal care; RE, Retinol equivalent; THC, Teda Health Center; VIF, Variance inflation factor; WHO, World Health Organization.

(AOR = 1.59, 95% CI; 1.09–2.32) and meal frequency of $\geq 4$ (AOR = 1.64, 95% CI; 1.09–2.32) were statistically significant.

## Conclusion and recommendation

The majority of respondents had inadequate consumption of foods rich in vitamin A. Educational status, family size, wealth index, dietary diversity, and meal frequency were found to be factors that affect adequate consumption of vitamin A-rich foods. Encouraging and educating lactating mothers to consume foods rich in vitamin A is crucial.

## Background

Vitamin A deficiency is a major public health problem, especially among low-income countries [1–3]. It is one of the leading causes of morbidity and mortality for those who have an inadequate intake of vitamin A from food [4]. Lactating mothers are the most vulnerable population in addition to children under 5 years old to vitamin A deficiency because they need a high amount to provide enough of vitamin A for their infant during lactation and to replace the loss that occurs through breastfeeding [5]. Generally, the recommended daily allowance of vitamin A for lactating mothers is 1300 µg/d [6].

Two forms of vitamin A are available in foods, namely preformed vitamin A (retinol) that is found in animal products such as human milk, glandular meats, liver and fish liver oils, egg yolk, whole milk, and other dairy products, and provitamin A (carotenoids) that is found in plant sources like green leafy vegetables (e.g. spinach and kale), yellow vegetables (e.g. pumpkins, squash, and carrots), yellow and orange non-citrus fruits (e.g. mangoes, apricots, and papayas [7].

Little attention has been given to the problem of vitamin A deficiency in breastfeeding as compared to preschool children because the clinical symptoms of xerophthalmia in women are rare [8]. Epidemiological studies revealed a high prevalence of low levels of serum retinol in lactating mothers in different Asian countries including 37% in Bangladesh [9], 13% in Bhaktapur, Nepal [10], and 18% in Indonesia [11]. Moreover, a high magnitude of vitamin A deficiency has been demonstrated in different African countries. For example, about 40% of lactating mothers were vitamin A deficient in Makhaza, Zimbabwe [12], 38% mothers in Zambia [13], and 10% of women were severely deficient in Kenya [14].

In Ethiopia, research has shown that vitamin A deficiency remains public health concern, including a high prevalence of Bitot's spots, with the highest level being in the Amhara region [15–17]. According to the Ethiopian national micronutrient survey report, the national prevalence of vitamin A deficiency among women of the reproductive age group was 3.4% [18].

Vitamin A is an essential fat-soluble vitamin required in a small amount for human beings for the optimal performance of different systems of our body including vision, reproduction, growth, and development, immunity, and integration of epithelium [1, 19]. Several causes of vitamin A deficiency in pregnant women have been identified, including inadequate consumption of vitamin A-rich foods, socio-demographic factors, economic status, duration of lactation (prolonged), amount of body fat, and hemoglobin concentration [2, 11, 12, 14, 20]. Vitamin A deficiency has a wide range of consequences, with the most common manifestations including ocular manifestations (e.g. xerophthalmia), mortality, failure to thrive, and poor reproduction [21]. In addition to the above consequences, vitamin A deficiency has an intergenerational effect. Children born with low vitamin A reserves and from deficient

mothers are at greater risk to develop vitamin A deficiency [8]. Attempts to prevent and control vitamin A deficiency include promoting vitamin A rich food consumption, providing vitamin A supplementation to lactating mothers and children, treatment of conditions and disease with vitamin A, and fortification of cooking oil with vitamin A [22].

In 2013, Ethiopia launched a revised national nutrition program which included strengthening of nutrition-sensitive intervention that mainly focused on promoting home gardening, community horticulture production, strengthen fruit and vegetable consumption, promoting urban gardening, increase access, and utilization of animal source foods as part of the strategic objective, which has made a great contribution to preventing vitamin A deficiency [23]. However, little is known regarding the consumption of vitamin A-rich food among lactating mothers. Hence, the present study was aimed at assessing the consumption of vitamin A-rich foods, and associated factors by lactating mothers.

## Methods

### Study design and setting

An institution-based cross-sectional study was conducted from February to March 2019 at health institutions in Gondar town. Gondar town is located in the North Gondar Zone of the Amhara Region and is 750 km away from Addis Ababa, the capital city of Ethiopia. According to the Gondar Town health office report, it is estimated that there are currently 11,412 pregnant and lactating mothers in the Town. There are 8 health centers and one comprehensive specialized referral hospital. The major food products are maize, sorghum, teff, bean, and other legumes. potato, spinach, lettuce, cabbage, kale, pumpkin, and fruits available at the study area include mango, avocado, banana and orange [24].

### Sample size calculation and sampling procedure

All lactating mothers visiting child immunization (EPI) and postnatal centers (PNC) at health institutions in Gondar Town were considered as source populations while those mothers in the selected health facilities were considered as study population. Lactating mothers visiting EPI and PNC in selected health facilities during the study period in Gondar Town were included in the study. The sample size was calculated based on single population proportion assumptions. Assuming 95% confidence level, 50% prevalence of adequate consumption of vitamin A-rich foods, a margin of error 5%, design effect of 2, 10% of non-response rate and applying population correction formula, the final sample size was 631. Study participants were selected by simple random sampling followed by a systematic sampling technique. From 9 health facilities, 5 health institutions (Gondar university comprehensive specialized referral hospital (GUCSRH), Maraki Health Center (MHC), Poly Health Center (PHC), Azezo Health Center (AHC), and Teda Health Center (THC), were selected by simple random sampling. By allocating samples proportionally, 179 from GUCSRH, 124 from PHC, 172 from MHC, 84 from AHC and 72 from THC were selected. The first participant was selected by lottery method and every other woman was included until the desired sample size was achieved.

### Operational definitions

The following operational definitions were used in this study. Adequate Vitamin A-rich foods consumed: Consumption of animal sources of vitamin A for > 4 days per week, or consumption of weighted source (total consumption of animal and vegetable sources of vitamin A) for > 6 days per week [25]. Minimum women dietary diversity: Consumption of at least 5 food groups from 10 food groups [26]. Poor nutritional knowledge: Individuals with scores

0–3 from 9 points were classified as having poor nutrition knowledge. Fair nutritional knowledge: Those with scores 4–6 were considered as having fair nutritional knowledge. Adequate nutritional knowledge: Those with scores of 7–9 were classified as having adequate nutritional knowledge [27].

## Data collection tool and procedure

The data were collected via interview using a structured questionnaire from Hellen Keller International Food Frequency Questionnaire (FFQ) after adapting to the local context for measuring the consumption of vitamin A-rich foods [25], and Food and Agriculture Organization (FAO) to measure woman's dietary diversity [26]. Consumption of vitamin A-rich food was assessed with food items listed in the Hellen Keller guideline. Those food items included for the analysis of vitamin A-rich food consumption include plant source foods (dark green leafy vegetables (DGLVs), carrot, pumpkin, mango, papaya, palm oil), and animal sources (egg, fish, liver, butter).

The FFQ asks respondents how many days in the last week they consumed the foods listed on a predesigned FFQ. Only major sources of vitamin A are taken into consideration (t100 RE), though some attention is given to other foods. The tool has 28 different food item questions. From the 28 food items, we used 21 food items which are available in the study area and consumed by the society, the remaining 7 food items like, noodles, Amaranth leaves, sweet potato leaves, cod liver oil, coconuts, weaning food fortified with vitamin A, Margarine fortified with vitamin A were not included in the tool because they are not commonly consumed in the study area. Maternal dietary diversity score was collected and calculated as the sum of the number of different food groups consumed by the mother with in 24 hrs preceding the survey. A total of ten food groups were considered in this study which includes; grains, pulses, beans and lentils, nuts and seeds, milk and dairy products, meat and fish, egg, DGLVs, other vitamin A-rich fruits and vegetables, other vegetables and other fruits. Ten data collectors who have a BSc degree in nursing and 5 supervisors with master's degrees participated in the data collection procedure.

## Data processing and analysis

Data was entered using Epi-info version 7 statistical software and exported to Stata version 14 for analysis. Descriptive and summary statistics were performed to describe the study population and were presented using tables and graphs. A principal component analysis was used to analyze the household wealth status of the respondents for urban and rural residents by considering their household assets and propertiesand it was categorized into the lowest, middle and highest tertiles. Factors associated with the consumption of vitamin A-rich food sources were identified by using a binary logistic regression model after checking the necessary assumptions of the model. Variables that fulfilled the Chi-square assumptions with a p-value of < 0.2 in the bivariable analysis were considered for multivariable logistic regression analysis. Those variables with p-value < 0.05 in the multivariable logistic regression analysis were declared as statistically significant. Both crude and adjusted odds ratios with the corresponding 95% confidence interval were calculated to measure the presence and strength of the association between adequate vitamin A consumption and contributing factors. Hosmer and Lemeshow goodness of fit test was conducted to test the model fitness and the model was adequate (p-value = 0.872). Multi-collinearity was checked by using variance inflation factor (VIF) and no multi-collinearity was found (VIF < 5).

### Data quality assurance

The consistency of the questionnaire was maintained by translating the English version to Amharic and then back to English. Before the actual data collection day, training was given for data collectors and supervisors. Pretest was conducted among 31 respondents at Ginbot 20 health center and necessary modification was made accordingly. The accuracy and completeness of the collected data were checked daily by the principal investigator. Moreover, the data were checked by inspection, and crosschecking the entered data with the questionnaire.

### Ethics approval and consent to participate

Ethical clearance was obtained from the Ethical Review Board of the University of Gondar. A letter of permission was obtained from the Gondar town health office and the respective medical director of each health facility under study. The objective of the study was explained and verbal consent was secured from the study participants. The right of participants to withdraw from the study at any time without any precondition was disclosed. Moreover, the confidentiality of the information obtained was guaranteed by all data collectors and investigators.

## Results

### Socio-demographic and economic characteristics

A total of 624 lactating mothers participated in the study giving a response rate of 98.89%. The median (median ± IQR) age of the respondents was 27 ± 6 years. The majority (84.13%) of mothers were urban residents. About 22.9% of mothers had an educational level of diplomas and above. Half of the study subjects had a household family size of ≤ 3 and around 33.2% of respondents were within the higher class in terms of economic status (Table 1).

### Maternal dietary diversity

From the ten food groups used for assessment of maternal dietary diversity, grains (75.48%) and legumes (62.07%) were highly consumed within 24 hours preceding the survey. The proportion of mothers who consumed ≥ 5 food groups within 24 hours preceding the survey was 43.75% (Table 2).

### Health service and nutritional advice related characteristics

The majority (92.8%) of the participants gave birth to their last child at the institution. About 66.7% of mothers had ANC visits 4 times and above during their last pregnancy (Table 3).

### Source of nutritional advice for lactating mothers

The majority of study participants received information on eating frequently during lactation from health professionals (Table 4).

### Nutritional knowledge

Nine questions were used to assess the nutritional knowledge of the respondents. The proportion of adequate nutritional knowledge was found to be 49.8% (Table 5).

### Vitamin A-rich food consumption

From animal sources of vitamin A, meat was consumed by 92.3% mothers with in the last week before the survey. Next to meat, milk was the most highly consumed animal source (69.72%). Among plant sources, palm oil (88.8%) and Dark green leafy vegetables (67.15%)

**Table 1. Socio-demographic and economic characteristics of lactating mothers visiting EPI and PNC at health institutions in Gondar Town, Northwest Ethiopia, 2019 (n = 624).**

| Variable | Category | Frequency | Percent (%) |
|---|---|---|---|
| **Residence** | Rural | 99 | 15.9 |
| | Urban | 525 | 84.1 |
| **Age group** | ≤ 24 years | 184 | 29.5 |
| | 25–34 years | 356 | 57.1 |
| | ≥ 35 years | 84 | 13.5 |
| **Religion** | Orthodox | 531 | 85.1 |
| | Muslim | 87 | 13.9 |
| | Others (protestant, catholic) | 6 | 1.0 |
| **Marital status** | Single | 28 | 4.5 |
| | Married | 552 | 88.5 |
| | Cohabiting | 26 | 4.2 |
| | Others (widowed, divorced) | 18 | 2.9 |
| **Educational status** | Unable to read and write | 109 | 17.5 |
| | Able to read and write | 115 | 18.4 |
| | Primary school | 118 | 18.9 |
| | Secondary school | 139 | 22.3 |
| | Diploma and above | 143 | 22.9 |
| **Occupation** | Government employee | 109 | 17.5 |
| | NGO employee | 15 | 2.4 |
| | Self- employed | 124 | 19.9 |
| | Housewife | 356 | 57.1 |
| | Others (student, daily laborer) | 20 | 3.2 |
| **Family size** | 1–3 | 312 | 50.0 |
| | 4–6 | 266 | 42.6 |
| | >6 | 46 | 7.4 |
| **Wealth index** | Lower class | 208 | 33.3 |
| | Middle class | 209 | 33.5 |
| | Higher class | 207 | 33.2 |

**Table 2. Food groups consumed 24 hours before the survey by lactating mothers visiting EPI and PNC at health institutions in Gondar Town, Northwest Ethiopia, 2019 (n = 624).**

| Food items | Frequency | Percent (%) |
|---|---|---|
| Grains | 471 | 75.48 |
| Pulses, beans, peas and lentils | 387 | 62.07 |
| Nuts and seeds | 115 | 18.43 |
| Milk and milk products | 369 | 59.13 |
| Meat, poultry and fish | 373 | 59.78 |
| Egg | 289 | 46.31 |
| Dark green leafy vegetables | 279 | 44.71 |
| Other vitamin A- rich fruits and Vegetables | 272 | 43.59 |
| Other vegetables | 187 | 29.97 |
| Other fruits | 212 | 33.97 |
| Dietary Diversity Score | | |
| < 5 food groups | 351 | 56.3 |
| ≥ 5 food groups | 273 | 43.7 |

**Table 3. Health service-related characteristics of lactating mothers visiting EPI and PNC at health institutions in Gondar Town, Northwest Ethiopia, 2019 (n = 624).**

| Variable | Frequency | Percent (%) |
|---|---|---|
| **Place of delivery** | | |
| Institution | 579 | 92.8 |
| Home | 45 | 7.2 |
| **ANC visit** | | |
| < 4 | 208 | 33.3 |
| ≥ 4 | 416 | 66.7 |
| **Nutritional counseling to eat more frequently during lactation** | | |
| Yes | 495 | 79.3 |
| No | 129 | 20.7 |
| **Nutritional counseling to eat animal sources frequently during lactation** | | |
| Yes | 472 | 75.6 |
| No | 152 | 24.4 |
| **Sources of information to eat fruit and vegetables frequently during lactation** | | |
| Yes | 460 | 73.7 |
| No | 164 | 26.3 |

were consumed relatively higher than other plant sources. Adequate consumption of vitamin A-rich food was 38.9% (95% CI: 35%- 43%) (Table 6).

## Factors associated with adequate consumption of vitamin A-rich food sources

The association between adequate consumption of vitamin A-rich foods and predictors was analyzed using binary logistic regression. All factors which fulfilled the chi-square assumptions and p-value <0.2 in univariable analysis were considered for the multivariable logistic regression model. Hence, residence, occupational status, educational status, household family size, wealth index, dietary diversity, meal frequency, place of delivery, ANC visit, nutritional counseling to eat more frequently during lactation, nutritional counseling to eat animal sources frequently during lactation, nutritional counseling to eat fruit and vegetables frequently during lactation and nutritional knowledge were included in the multivariable analysis. Educational status, household family size, wealth index, dietary diversity and meal frequency were found to be statistically significant at p-value <0.05 and were considered to be determinants for adequate consumption of vitamin A-rich food sources.

The odds of eating adequate vitamin A-rich foods among mothers whose educational level is college and above was 2.28 times higher as compared to those unable to read and write (AOR = 2.26, 95% CI; 1.02–4.99). Those mothers with a household family size of ≤ 3 had 4 times more likely chance of adequate consumption than those with family size more than 6 (AOR = 4.04, 95% CI; 1.60–10.17). Household economic status was found to be a significant predictor of adequate consumption of vitamin A food sources. Mothers who were in the higher economic class had 1.93 times more likely chance of consuming vitamin A-rich foods adequately as compared to those who belong to lower economic class (AOR = 1.93, 95% CI; 1.18–3.14). Furthermore, the odds of adequate consumption among mothers who had a dietary diversity score of ≥ 5 were 1.59 times higher than their counterparts (AOR = 1.59, 95% CI; 1.09–2.32). The consumption of vitamin A-rich food was found to be 1.61 times higher for those mothers who had > 3 meals per day as compared with those who had smaller meal frequency (AOR = 1.61, 95%CI; 1.09–2.32) (Table 7).

**Table 4. Sources of nutritional advice or for lactating mothers visiting EPI and PNC at health institutions in Gondar Town, Northwest Ethiopia, 2019.**

| Source of information | Frequency | Percent (%) |
|---|---|---|
| **To eat more frequently during lactation (n = 495)** | | |
| Health professional | | |
| Yes | 425 | 85.9 |
| No | 70 | 14.1 |
| Health extension workers | | |
| Yes | 31 | 6.3 |
| No | 464 | 93.7 |
| Family /friends | | |
| Yes | 59 | 11.9 |
| No | 436 | 90.5 |
| Mass media | | |
| Yes | 4 | 0.8 |
| No | 491 | 99.2 |
| **To eat animal sources frequently during lactation (n = 472)** | | |
| Health professional | | |
| Yes | 398 | 84.3 |
| No | 74 | 15.7 |
| Health extension workers | | |
| Yes | 26 | 5.5 |
| No | 446 | 94.5 |
| Family /friends | | |
| Yes | 51 | 10.8 |
| No | 421 | 89.2 |
| Mass media | | |
| Yes | 18 | 3.8 |
| No | 454 | 96.2 |
| **To eat fruit and vegetables frequently during lactation (n = 460)** | | |
| Health professional | | |
| Yes | 385 | 83.3 |
| No | 77 | 16.7 |
| Health extension workers | | |
| Yes | 33 | 7.2 |
| No | 427 | 92.8 |
| Family /friends | | |
| Yes | 46 | 10 |
| No | 414 | 90 |
| Mass media | | |
| Yes | 28 | 6.1 |
| No | 432 | 93.9 |

## Discussion

Our findings showed that the consumption of adequate vitamin A-rich foods was 38.9% (95% CI: 35%- 43%). Lactating women have a higher vitamin A requirement compared to women who are not lactating as vitamin A of the mother is directly transferred into breast milk. As such, the FAO/WHO recommends an extra 350 RE per day throughout lactation. These recommendations are based generally on the expected secretion of retinol into human milk. Since

**Table 5. Nutritional knowledge of lactating mothers visiting EPI and PNC at health institutions in Gondar Town, Northwest Ethiopia, 2019(n = 624).**

| Question | Frequency | Percent (%) |
|---|---|---|
| **Meaning of nutrition** | | |
| Related to the importance of food to the body | 436 | 69.9 |
| I don't know | 188 | 30.1 |
| **Reason for eating** | | |
| For growth and development | 402 | 64.4 |
| To satisfy hunger | 222 | 35.6 |
| **Description of a balanced diet** | | |
| Diet containing an appropriate amount of each nutrient | 390 | 62.5 |
| Diet containing all the necessary nutrients | 234 | 37.5 |
| **Presence of disease due to inadequate intake of foods** | | |
| Yes | 565 | 90.5 |
| No | 59 | 9.5 |
| **Energy giving foods** | | |
| Starchy foods | 292 | 46.8 |
| Fruits and vegetables | 332 | 53.2 |
| **Body- building foods** | | |
| Animal foods and legumes | 412 | 66.0 |
| Fruits, fats, and oils | 212 | 34.0 |
| **Protective foods** | | |
| Fruits and vegetables | 345 | 55.3 |
| Starchy roots, plantains, and oils | 279 | 44.7 |
| **Importance of snack for HIV infected person** | | |
| Yes | 530 | 84.9 |
| No | 94 | 15.1 |
| **Importance of exercise to human** | | |
| Yes | 600 | 96.2 |
| No | 24 | 3.8 |
| **Nutritional knowledge** | | |
| Poor | 64 | 10.3 |
| Fair | 249 | 39.9 |
| Adequate | 311 | 49.8 |

the concentration of vitamin A in human milk is dependent on the mothers' status, their infants would also be expected to benefit [28]. Our results are higher than those found in Damot Sore (Southern Ethiopia) and rural Kenya. Indeed, in the Damot Sore district, the study revealed that only 12.5% of pregnant mothers had consumed either animal or plant source food more than 3 times per week. While in India, the study revealed that 20% of the study participants had adequate vitamin A-rich food intake [14, 29, 30]. The difference in results can be explained by the effect of sample size, study settings, and physiological state (being pregnant may reduce intake by decreasing appetite). Furthermore, the present study had higher findings than a study done in Bangladesh which found only 13% of lactating mothers had intakes of vitamin A above the recommended dietary allowance [9]. This difference may reflect differing economic status, as the study participants in Bangladesh were poor urban mothers and their economic status was a major determinant for adequate consumption of foods rich in vitamin A.

**Table 6. Vitamin A-rich food groups consumed at least once during last week by lactating mothers visiting EPI and PNC at health institutions in Gondar Town, Northwest Ethiopia, 2019 (n = 624).**

| Food groups | Frequency | Percent (%) |
|---|---|---|
| Meat | 576 | 92.3 |
| Milk | 435 | 69.7 |
| Egg | 378 | 60.6 |
| Chicken | 445 | 71.3 |
| Fish | 49 | 7.9 |
| Liver | 111 | 17.8 |
| Butter | 394 | 63.1 |
| Rice | 317 | 50.8 |
| Hot peppers | 458 | 73.4 |
| DGLVs | 419 | 67.2 |
| Carrot | 226 | 36.2 |
| Mango | 189 | 30.3 |
| Pumpkin | 95 | 15.2 |
| Papaya | 132 | 21.2 |
| Spinach | 326 | 52.2 |
| Peanut | 271 | 43.4 |
| Sweet potato | 128 | 20.5 |
| Lentil or others legumes | 544 | 87.2 |
| Palm oil | 613 | 98.2 |
| Apricot | 121 | 19.4 |

A high level of adequate vitamin A-rich food consumption was demonstrated among mothers with educational level of college and above compared to those who are illiterate. This was supported by a study done in Nepal which showed that there was a positive relationship between adequate micronutrient intake and the mother's educational level [5]. This is because mothers who attended college and above are more likely to get more information about vitamin A food sources and the consequence of inadequate dietary vitamin A intake. In this study, mothers with a family size of ≤ 3 had 4 times more likely chance of consuming adequate vitamin A-rich foods as compared to those with > 6 family size. This may be due to a difficulty to provide adequate foods to large family and mothers may sacrify their quality/quantity of nutrition to protect their children. Mothers with upper-class wealth index had 1.93 times more likely chance to consume adequate vitamin A-rich foods compared to the lower class. This may be because mothers with low socioeconomic status may not have access or cannot afford to buy food rich in vitamin A. The odds of adequate consumption of vitamin A among mothers with a dietary diversity score of ≥ 5 were 1.59 times higher than their counterparts. This can be supported by the study in Nepal which found that a positive relation between micronutrient consumption and dietary diversity [5]. According to WHO guidelines, a child who consumed foods from at least 4 food groups on a preceding day has a high possibility of consumption of at least one animal-source food and one fruit or vegetable on that day, in addition to a staple food such as grain in most populations [31]. Furthermore, adequate consumption of vitamin A-rich food was associated with the number of meals within a day. Those mothers who had > 3 meals per day were 1.61 times more likely to consume adequate vitamin A-rich foods as compared to those who had ≤ 3 meals. This may be because as the number of meals increases it is more likely to increase the variety of foods and foods containing vitamin A.

**Table 7. Bivariable and multivariable logistic regression model predicting the likelihood of adequate consumption of vitamin A-rich food sources by lactating mothers visiting EPI and PNC at health facilities in Gondar Town, Northwest Ethiopia, 2019.**

| Variable | Vitamin A- rich food consumption | | COR (95%CI) | AOR (95%CI) |
|---|---|---|---|---|
| | Adequate | Inadequate | | |
| **Residence** | | | | |
| Rural | 27 | 72 | 1 | 1 |
| Urban | 216 | 309 | 1.86 (1.16–2.99) | 0.79(0.41–1.51) |
| **Occupation** | | | | |
| Housewife | 112 | 244 | 1 | 1 |
| Government employee | 62 | 47 | 2.87 (1.85–4.46) | 0.92 (0.48–1.77) |
| NGO employed | 7 | 8 | 1.91 (0.67–5.39) | 0.86 (0.25–2.95) |
| Self- employed | 55 | 69 | 1.74 (1.14–2.64) | 1.22 (0.76–1.95) |
| Others | 7 | 13 | 1.17 (0.46–3.02) | 0.92 (0.32–2.63) |
| **Educational status** | | | | |
| Unable to read and write | 30 | 79 | 1 | 1 |
| Able to read and write | 33 | 82 | 1.06 (0.59–1.90) | 0.73 (0.37–1.42) |
| Primary school | 40 | 78 | 1.35 (0.77–2.38) | 0.88 (0.45–1.73) |
| Secondary school | 48 | 91 | 1.39 (0.80–2.40) | 0.75 (0.38–1.49) |
| College and above | 92 | 51 | 4.75 (2.76–8.17) | 2.26 (1.02–4.99)* |
| **Family size** | | | | |
| >6 | 7 | 39 | 1 | 1 |
| 4–6 | 107 | 159 | 3.94 (1.70–9.10) | 3.06 (1.03–7.64)* |
| 1–3 | 129 | 183 | 3.75 (1.62–8.69) | 4.04 (1.60–10.17)* |
| **Wealth index** | | | | |
| Lower class | 55 | 153 | 1 | 1 |
| Middle class | 74 | 135 | 1.52 (1.00–2.31) | 1.11 (0.69–1.76) |
| Upper class | 114 | 93 | 3.41 (2.26–5.15) | 1.93 (1.18–3.14)* |
| **Dietary diversity** | | | | |
| < 5 food groups | 114 | 237 | 1 | 1 |
| ≥ 5 food groups | 129 | 144 | 1.86 (1.34–2.58) | 1.59 (1.09–2.32)* |
| **Meal frequency** | | | | |
| ≤ 3 times | 129 | 280 | 1 | 1 |
| > 3 times | 114 | 101 | 2.45 (1.74–3.44) | 1.61 (1.09–2.32)* |
| **Place of delivery** | | | | |
| Home | 6 | 39 | 1 | 1 |
| Institution | 237 | 342 | 4.50 (1.88–10.81) | 2.03(0.77–5.31) |
| **ANC visit** | | | | |
| < 4 visits | 55 | 153 | 1 | 1 |
| ≥ 4 visits | 188 | 228 | 2.29 (1.59–3.29) | 1.36 (0.87–2.12) |
| **Nutritional counseling to eat frequently during lactation** | | | | |
| No | 30 | 99 | 1 | 1 |
| Yes | 213 | 282 | 2.49 (1.59–3.89) | 0.91 (0.44–1.87) |
| **Nutritional counseling to eat animal during lactation** | | | | |
| No | 36 | 116 | 1 | 1 |
| Yes | 207 | 265 | 2.52 (1.66–3.81) | 1.01 (0.49–2,04) |
| **Nutritional counseling to eat fruit and vegetables during lactation** | | | | |
| No | 37 | 127 | 1 | 1 |
| Yes | 206 | 254 | 2.78 (1.85–4.19) | 1.46 (0.77–2.76) |
| **Nutritional knowledge** | | | | |

(*Continued*)

**Table 7.** (Continued)

| Variable | Vitamin A- rich food consumption | | COR (95%CI) | AOR (95%CI) |
|---|---|---|---|---|
| | Adequate | Inadequate | | |
| Poor | 15 | 49 | 1 | 1 |
| Fair | 79 | 170 | 1.52 (0.80–2.87) | 1.08 (0.51–2.29) |
| Adequate | 149 | 162 | 3.00 (1.62–5.58) | 1.71 (0.79–3.68) |

* = p-value <0.05, COR = crude odds ratio and AOR = adjusted odds ratio.

## Limitations of the study

Because the study was institution-based, it may overestimate consumption and may not be representative of the community as those who come to health facilities may have relatively good consumption since they may have some information regarding vitamin A-rich foods and the possible consequences of inadequate intake. The study may be prone to both recall bias and social desirability bias as the tool assessed the intake of food before a week. Due to the lack of previous studies for comparison, we used community-based studies and different population groups. Serum retinol levels were not determined, due to a lack of required facilities.

## Conclusion

Based on the Hellen Keller threshold values, 61.1% of lactating mothers have inadequate vitamin A-rich food consumption and thus, was not a public health problem among our population. Having a college education level and above, household family size of 3 and below, higher economic class, having minimum dietary diversity score and meal frequency of 4 and above were predictors for adequate consumption of vitamin A-rich food sources. Lactating mothers and the community as a whole should be educated on the importance of consuming locally available foods rich in vitamin A. Furthermore, the government should strengthen various strategies to be used nationally in the prevention and control of micronutrient deficiencies in Ethiopia.

## Supporting information

**S1 Data.**
(DTA)

## Acknowledgments

We are indebted to the study participants, data collectors and supervisors for their time and commitment to take part in the study.

## Author Contributions

**Conceptualization:** Addisalem Damtie Aserese, Azeb Atenafu, Mekonnen Sisay, Muluken Bekele Sorrie, Birhanu Wubale Yirdaw, Martha Kassahun Zegeye.

**Data curation:** Addisalem Damtie Aserese, Azeb Atenafu, Mekonnen Sisay, Muluken Bekele Sorrie, Birhanu Wubale Yirdaw, Martha Kassahun Zegeye.

**Formal analysis:** Addisalem Damtie Aserese, Azeb Atenafu, Mekonnen Sisay, Muluken Bekele Sorrie, Birhanu Wubale Yirdaw, Martha Kassahun Zegeye.

**Investigation:** Addisalem Damtie Aserese, Azeb Atenafu, Muluken Bekele Sorrie.

**Methodology:** Addisalem Damtie Aserese, Azeb Atenafu, Mekonnen Sisay, Muluken Bekele Sorrie, Birhanu Wubale Yirdaw, Martha Kassahun Zegeye.

**Software:** Addisalem Damtie Aserese, Azeb Atenafu, Mekonnen Sisay, Muluken Bekele Sorrie.

**Supervision:** Addisalem Damtie Aserese, Azeb Atenafu, Mekonnen Sisay, Muluken Bekele Sorrie, Birhanu Wubale Yirdaw.

**Validation:** Addisalem Damtie Aserese.

**Writing – original draft:** Addisalem Damtie Aserese, Azeb Atenafu, Mekonnen Sisay, Muluken Bekele Sorrie, Birhanu Wubale Yirdaw, Martha Kassahun Zegeye.

**Writing – review & editing:** Addisalem Damtie Aserese, Azeb Atenafu, Mekonnen Sisay, Muluken Bekele Sorrie, Birhanu Wubale Yirdaw, Martha Kassahun Zegeye.

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
