## [Decision Letter · Decision Letter 0]

19 Nov 2019

PONE-D-19-28467

Adequate vitamin A rich food consumption and associated factors among lactating
mothers visiting child immunization and post-natal clinic at health institutions in
Gondar Town, Northwest Ethiopia

PLOS ONE

Dear Addisalem Damtie,

Thank you for submitting your manuscript to PLOS ONE. After careful consideration, we
feel that it has merit but does not fully meet PLOS ONE’s publication criteria as it
currently stands. Therefore, we invite you to submit a revised version of the
manuscript that addresses the points raised during the review process.

According to many literature Vitamin A play a pivotal role for grow and
differentiation of cell and tissue. It has extra benefit in the healthy development
of fetus and neonates. I would like to commend authors for conducting this very
important topic. 

Abstract:

The section seems well written has precisely and briefly illustrated the entire
study.

Introduction:

This section has shown the scientific definition of vitamin A, the epidemiology of
vitamin A, the prevalence and the vulnerable groups. However, the writing looks
juggling from one topic to the other without clear coherence. It lacks consistency.
There is not enough literature included to expound the objective. May be, good to
incorporate literature that has been researched the topic at a national, regional
and local level.

Methods:

I don’t see the relevance in this study to write down detail information about the
study area. I recommend to make it brief and short with limited words.

There are many similar studies has been conducted in various part of Ethiopia, Why
you assumed 50 % prevalence of vitamin A rich food consumption to calculate the
sample size? What is your rationale to implement a designed effect? What is your
inclusion and exclusion criteria in this study?

Result:

This section appears to have incorporated information to address the objective and
the analysis seems sound. However, there are few sporadic technical issues in the
interpretation which has to be addressed accordingly.

Discussion:

This section looks very shallow and failed to make strong argument. As I have
indicated in the introduction section, there is no sufficient literature that
supposed to provide scientific clarification to each statistical analysis. Most of
the argument has been made based on assumption which makes the argument very frail
and less scientific. Your argument must be evidence based or supported by other
literature.  Oh the other hand, you have to address each objective included in this
study in the discussion section. Each analyzed results should be properly discussed.
Similarly, at the limitation of this study you indicated” the study was institution
based, it may fail to show the consumption of specific community” What do you mean
“consumption of specific community”? This and other similar statements needs your
attention to properly articulate pass the right message.  Overall, this section
needs extensive overhaul.

We would appreciate receiving your revised manuscript by Jan 03 2020 11:59PM. When
you are ready to submit your revision, log on to https://www.editorialmanager.com/pone/ and select the 'Submissions
Needing Revision' folder to locate your manuscript file.

If you would like to make changes to your financial disclosure, please include your
updated statement in your cover letter.

To enhance the reproducibility of your results, we recommend that if applicable you
deposit your laboratory protocols in protocols.io, where a protocol can be assigned
its own identifier (DOI) such that it can be cited independently in the future. For
instructions see: http://journals.plos.org/plosone/s/submission-guidelines#loc-laboratory-protocols

We look forward to receiving your revised manuscript.

Kind regards,

Solomon Assefa Woreta

Academic Editor

PLOS ONE

Journal Requirements:

2. Your ethics statement must appear in the Methods section of your manuscript. If
your ethics statement is written in any section besides the Methods, please move it
to the Methods section and delete it from any other section. Please also ensure that
your ethics statement is included in your manuscript, as the ethics section of your
online submission will not be published alongside your manuscript.

4. Please provide additional details regarding participant consent. In the ethics
statement in the Methods and online submission information, please ensure that you
have specified (1) whether consent was suitably informed and (2) what type you
obtained (for instance, written or verbal). If your study included minors under age
18, state whether you obtained consent from parents or guardians. If the need for
consent was waived by the ethics committee, please include this information.

Additional Editor Comments:

According to many literatures Vitamin A play a pivotal role for grow and
differentiation of cell and tissue. It has extra benefit in the healthy development
of fetus and neonates. I would like to commend authors for conducting this very
important topic.

Abstract:

The section seems well written has precisely and briefly illustrated the entire
study.

Introduction:

This section has shown the scientific definition of vitamin A, the epidemiology of
vitamin A, the prevalence and the vulnerable groups. However, the writing looks
juggling from one topic to the other without clear coherence. It lacks consistency.
There is not enough literature included to expound the objective. May be, good to
incorporate literatures that has been researched the topic at a national, regional
and local level.

Methods:

I don’t see the relevance in this study to write down detail information about the
study area. I recommend to make it brief and short with limited words.

There are many similar studies has been conducted in various part of Ethiopia, Why
you assumed 50 % prevalence of vitamin A rich food consumption to calculate the
sample size? What is your rationale to implement a designed effect? What is your
inclusion and exclusion criteria in this study?

Result:

This section appears to have incorporated information to address the objective and
the analysis seems sound. However, there are few sporadic technical issues in the
interpretation which has to be addressed accordingly.

Discussion:

This section looks very shallow and failed to make strong argument. As I have
indicated in the introduction section, there is no sufficient literature that
supposed to provide scientific clarification to each statistical analysis. Most of
the argument has been made based on assumption which makes the argument very frail
and less scientific. Your argument must be evidence based or supported by other
literature. Oh the other hand, you have to address each objective included in this
study in the discussion section. Each analyzed results should be properly discussed.
Similarly, at the limitation of this study you indicated” the study was institution
based, it may fail to show the consumption of specific community” What do you mean
“consumption of specific community”? This and other similar statements needs your
attention to properly articulate pass the right message. Overall, this section needs
extensive overhaul.

Reviewers' comments:

Reviewer's Responses to Questions

**Comments to the Author**

1. Is the manuscript technically sound, and do the data support the conclusions?

Reviewer #1: Yes

2. Has the statistical analysis been performed
appropriately and rigorously? 

Reviewer #1: Yes

3. Have the authors made all data underlying the
findings in their manuscript fully available?

Reviewer #1: Yes

4. Is the manuscript presented in an intelligible
fashion and written in standard English?

Reviewer #1: No

5. Review Comments to the Author

Reviewer #1: I have read an article titled “Adequate vitamin A rich food consumption
and associated factors among lactating mothers visiting child immunization and
post-natal clinic at health institutions in Gondar Town, Northwest Ethiopia” by
Damtie and collaborators with great interest. The article can help policy makers and
researchers who wish to work on Vitamin A. I have some issues the authors may wish
to consider before publication.

1. The English language needs extensive revision (detailed suggestions can be found
with the attachment).

2. Line 56: Using abbreviation in abstract is not recommended. This abbreviation
appeared for the first time here, and abbreviations should be written in full form
when they appear for the first time

3. Line 134: As "VAD" appears for the first time it is good if it is written in full
form. In general as this abbreviation only occurs at once in the entire manuscript,
it is good to use in its full form.

4. Line 143: "strengthen fruit and vegetable" is not clear.

5. Lines 150-160: citation is needed.

6. Line 168: It is not clear why authors used 50% prevalence while they have previous
study results. Of course those are community-based studies and may not apply for
institutional-based study like this one. However authors have used them to compare
the prevalence with current study. Authors may specify reason of using p-value of
50% and include lack of previous studies for comparison (if that is the case) as a
limitation.

7. Lines 192-193: Authors mentioned that pretested questionnaire was developed from
different sources mainly Hellen Keller International Food Frequency Questionnaire.
Authors are encouraged to cite at least some of these sources.

8. Lines 193-194: Authors mentioned that the HK-FFQ was adapted to the local context.
Did that mean the tool is validated? If so, the tool validation result may be cited
or at least explicitly mentioned.

Lines 213 to 214: Authors mentioned that the tool was pretested and necessary
modifications made. What modification were actually performed? What was the
reliability of nutritional knowledge questions from the pretest result.

9. Lines 220 to 221 is a repetition from line 205.

10. Lines 230 to 231: Variance inflation factor is more suitable for linear
regression and authors may use standard error to check co-linearity in binary
logistic regression.

11. Line 332: Authors should check the reference. The who report can not be cited for
a study in India.

12. Line 336: Authors used both full form and abbreviation RDA twice both at line 105
and here. What is the need of abbreviating recommended dietary allowance if the full
form is always written?

13. Line 368: This statement should be rewritten in a meaningful manner.

14. Lines 371 to 372: This statement should be reworked as assessment of intake of
food before a week seems reason of social desirability. I suggest the authors to
split the statement in to two.

15. Line 389: Authors may include the review protocol number.

16. Line 391: Authors may wish to disclose whether the consent obtained was written
or verbal.

Finally, the discussion seems straight jacketed. Authors may include additional
references and suggest additional justification. Authors may also discuss factors
which are not associated in the current study but were identified by previous
studies. Factors which are not associated are also important.

Good luck!

6. PLOS authors have the option to publish the peer
review history of their article (what does this mean?). If published, this will
include your full peer review and any attached files.

If you choose “no”, your identity will remain anonymous but your review may still be
made public.

**Do you want your identity to be public for this peer review?** For
information about this choice, including consent withdrawal, please see our
Privacy Policy.

Reviewer #1: Yes: Henok Dagne Derso

---

## [Author Response · Author response to Decision Letter 0]

10 Jan 2020

Dear Editor and Reviewer 

Sincerest thanks for your response and reviewers comments on our manuscript. We have
modified the paper in response to the extensive and insightful editor and reviewer
comments. We would be glad to respond to any further questions and comments that you
may have.

Response to Editor

1. Is the manuscript presented in an intelligible fashion and written in standard
English?

 We tried to amend the English according to the comment

2. Line 168: It is not clear why authors used 50% prevalence while they have previous
study results. Of course those are community-based studies and may not apply for
institutional-based study like this one. However authors have used them to compare
the prevalence with current study. Authors may specify reason of using p-value of
50% and include lack of previous studies for comparison (if that is the case) as a
limitation

• We used 50% prevalence to get the maximum sample size, since there is no study
conducted among lactating mothers, we put it as a limitation

3. What is your rationale to implement a designed effect? What are your inclusion and
exclusion criteria in this study?

• We used design effect of 2 to consider the loss of effectiveness by the use of
cluster sampling, instead of simple random sampling.

• The inclusion criteria were lactating mothers visiting child immunization (EPI) and
postnatal care (PNC) centers in selected health facilities during the study period
in Gondar Town.

• The exclusion criteria were lactating mothers who were severely ill during data
collection period; we didn’t get such type of respondent.

• We used median with IQR since the data is not normally distributed

4. The English language needs extensive revision (detailed suggestions can be found
with the attachment).

• We have extensively done the copy edit for our language usage. 

5. Line 56: Using abbreviation in abstract is not recommended. This abbreviation
appeared for the first time here, and abbreviations should be written in full form
when they appear for the first time

• It is modified accordingly to Helen Keller International Food Frequency
Questionnaire

6. Line 134: As "VAD" appears for the first time it is good if it is written in full
form. In general as this abbreviation only occurs at once in the entire manuscript,
it is good to use in its full form.

• It is modified accordingly to vitamin A deficiency

7. Line 143: "strengthen fruit and vegetable" is not clear.

• This was to mean strengthen fruit and vegetable consumption, and corrected
accordingly

8. Lines 150-160: citation is needed.

• Thank you for the comment, we have provided the citation

9. Line 168: It is not clear why authors used 50% prevalence while they have previous
study results. Of course those are community-based studies and may not apply for
institutional-based study like this one. However authors have used them to compare
the prevalence with current study. Authors may specify reason of using p-value of
50% and include lack of previous studies for comparison (if that is the case) as a
limitation.

• We used 50% prevalence to get the maximum sample size, since there is no study
conducted among lactating mothers; we put it as a limitation for using in the
discussion part

10. Lines 192-193: Authors mentioned that pretested questionnaire was developed from
different sources mainly Hellen Keller International Food Frequency Questionnaire.
Authors are encouraged to cite at least some of these sources.

• We have cited the references accordingly; when we say different sources it is to
mean that references like Food and Agriculture Organization (FAO) to measure woman’s
dietary diversity

11. Lines 193-194: Authors mentioned that the HK-FFQ was adapted to the local
context. Did that mean the tool is validated? If so, the tool validation result may
be cited or at least explicitly mentioned.

• For this study we didn’t do tool validation, since the HK-FFQ tool is
internationally accepted tool for assessing vitamin A rich food intakes, the tool
have 28 different food item questions. From the 28 food items we used 21 food items
which are available in the study area and consumed by the society, the remaining 7
food items like, noodles, Amaranth leaves, sweet potato leaves, cod liver oil,
coconuts, weaning food fortified with vitamin A, Margarine fortified with vitamin A
were not included in the tool because they are not commonly consumed in the study
area. Accordingly food frequency questionnaires can be adjusted to the local
context.

12. Lines 213 to 214: Authors mentioned that the tool was pretested and necessary
modifications made. What modifications were actually performed? What was the
reliability of nutritional knowledge questions from the pretest result?

• The modifications were mainly editorial like language translation of food items
from English to Amharic. We didn’t perform reliability test for the nutritional
knowledge questions, since it is the commonly used tool for assessing the
nutritional knowledge of mothers

13. A line 220 to 221 is a repetition from line 205.

• Modified accordingly

14. Lines 230 to 231: Variance inflation factor is more suitable for linear
regression and authors may use standard error to check co-linearity in binary
logistic regression.

• Of course standard error is also used for checking multicollinearity, in our case
standard error is also less than 4, and moreover, multicollinearity is a state of
very high inter-correlations or inter-associations among the independent variables.
It is therefore multicollinearity can also be detected with the help of tolerance
and its reciprocal, called variance inflation factor (VIF).

15. Line 332: Authors should check the reference. The WHO report cannot be cited for
a study in India.

• Checked and modified accordingly

16. Line 336: Authors used both full form and abbreviation RDA twice both at line 105
and here. What is the need of abbreviating recommended dietary allowance if the full
form is always written?

• Modified accordingly

17. Line 368: This statement should be rewritten in a meaningful manner.

• Modified to consumption of specific community

18. Lines 371 to 372: This statement should be reworked as assessment of intake of
food before a week seems reason of social desirability. I suggest the authors to
split the statement in to two.

• Noted and modified

19. Line 389: Authors may include the review protocol number.

• The protocol number for Ethical review was IPH/180/06/2011

20. Line 391: Authors may wish to disclose whether the consent obtained was written
or verbal.

• The consent obtained was verbal and included in the document

21. Finally, the discussion seems straight jacketed. Authors may include additional
references and suggest additional justification. Authors may also discuss factors
which are not associated in the current study but were identified by previous
studies. Factors which are not associated are also important

• Some amendments were made for the discussion part, the main problem was lack of
literatures

for editor and reviewers.docx
---

## [Decision Letter · Decision Letter 1]

18 Jun 2020

PONE-D-19-28467R1

Adequate vitamin A rich food consumption and associated factors among lactating
mothers visiting child immunization and post-natal clinic at health institutions in
Gondar Town, Northwest Ethiopia

PLOS ONE

Dear Dr. Aserese,

Thank you for submitting your manuscript to PLOS ONE. After careful consideration, we
feel that it has merit but does not fully meet PLOS ONE’s publication criteria as it
currently stands. Therefore, we invite you to submit a revised version of the
manuscript that addresses the points raised during the review process.

Apologies for the delay in returning this manuscript to you. The global pandemic has
delayed the review process. As you will see from the reviewer's comments below,
while one reviewer is satisfied with your revised manuscript the other still has
some concerns. I would add that I am personally not satisfied with the quality of
the written English in the manuscript and strongly encourage you to get a copy
editor to improve it. One of the journal's publication criteria requires the
manuscript to be written intelligibly and in standard English. Please keep this in
mind as you revise the manuscript. 

Please submit your revised manuscript by Aug 02 2020 11:59PM. If you will need more
time than this to complete your revisions, please reply to this message or contact
the journal office at plosone@plos.org. When
you're ready to submit your revision, log on to https://www.editorialmanager.com/pone/ and select the 'Submissions
Needing Revision' folder to locate your manuscript file.

If you would like to make changes to your financial disclosure, please include your
updated statement in your cover letter. Guidelines for resubmitting your figure
files are available below the reviewer comments at the end of this letter.

We look forward to receiving your revised manuscript.

Kind regards,

Robin D Clugston, Ph.D.

Academic Editor

PLOS ONE

Reviewers' comments:

Reviewer's Responses to Questions

**Comments to the Author**

1. If the authors have adequately addressed your comments raised in a previous round
of review and you feel that this manuscript is now acceptable for publication, you
may indicate that here to bypass the “Comments to the Author” section, enter your
conflict of interest statement in the “Confidential to Editor” section, and submit
your "Accept" recommendation.

Reviewer #1: All comments have been addressed

Reviewer #2: All comments have been addressed

2. Is the manuscript technically sound, and do the data
support the conclusions?

Reviewer #1: Yes

Reviewer #2: Partly

3. Has the statistical analysis been performed
appropriately and rigorously? 

Reviewer #1: Yes

Reviewer #2: I Don't Know

4. Have the authors made all data underlying the
findings in their manuscript fully available?

Reviewer #1: (No Response)

Reviewer #2: Yes

5. Is the manuscript presented in an intelligible
fashion and written in standard English?

Reviewer #1: Yes

Reviewer #2: No

6. Review Comments to the Author

Reviewer #1: Authors fully addressed my comments in the revised manuscript. They have
improved the write up and addressed major issues raised.

Reviewer #2: authors have clarified several of the questions raised in the previous
review, unfortunatly, the major problem in this paper is the discussion part (which
was stated by the previous reviewer) that authors can overcome by including more
references, and discussing their main findings related to their results and study
population. the discussion can be improved, be more organized and enriched.

Abstract

(line 54) : can authors clarify ... the authors stated that about 631 study
participants were included ... and in line 61: a total of 624 of lactating women
participated... can author give the reason of the loss (-7)

Background

Needs to be organized… I suggest to the authors to start the first paragraph by line
111…117

Line 112 : not only vitamin A deficiency is the leading cause of morbidity and
mortality, also zinc deficiency and both are the most leading causes…

Line 113 : … vulnerable population segment by for those who have inadequate intake of
vitamin A from food

Line 114 : lactating women are the most… in addition to children under 5y old.

Line 115 : … amount of micronutrients… I thnik here you have to put enough of vit
A

Line 118… line 122 can be deleted

Line 129 : « …remains… » is there any measures taken nationnally ?

Line 132 : … of reproductive age… what about lactating women ? is there any regional
or national statistics ?

Authors can reinsert here paragraph starting by line 98 before line 134

Line 138 to line 144 , this paragraph needs edits and rephrasing (children who born
with low vitA reserves and from defiscient mothers are at greater risk to develop a
vitA deficniency

Line 202 : adapted questionnaire instead of pretested questionnaire

Line 206 : .. and other litteratures, can the authors be more precise

I suggest to authors that data quality assurance comes after data analysis

Line 226 running frequency ????

Line 250 was obtained instead of secured

Table 1 did authors collect information about household income ??

I think that the discussion needs a more in-depth , and authors have to present their
major findings related to the population studied and not children aged between 6 to
23 months… to compare them to other studies conducted among lactating women (and
explain in details the results obtained)

Authors may add more statements in the end to highlight the implication and
conclusions from this study

7. PLOS authors have the option to publish the peer
review history of their article (what does this mean?). If published, this will
include your full peer review and any attached files.

If you choose “no”, your identity will remain anonymous but your review may still be
made public.

**Do you want your identity to be public for this peer review?** For
information about this choice, including consent withdrawal, please see our
Privacy Policy.

Reviewer #1: Yes: Henok Dagne Derso

Reviewer #2: No

---

## [Author Response · Author response to Decision Letter 1]

27 Jun 2020

Dear Editor and Reviewer 

Sincerest thanks for your response and reviewers comments on our manuscript. We have
modified the paper in response to the extensive and insightful editor and reviewer
comments. We would be glad to respond to any further questions and comments that you
may have.

Response to Editor and Reviewer

1. Abstract

(line 54) : can authors clarify ... the authors stated that about 631 study
participants were included ... and in line 61: a total of 624 of lactating women
participated... can author give the reason of the loss (-7)

Seven lactating women discontinue the interview after starting due to their personal
reasons

2. Background Needs to be organized… I suggest to the authors to start the first
paragraph by line 111…117

Thank you for the suggestion and we have modified accordingly

3. Line 112 : not only vitamin A deficiency is the leading cause of morbidity and
mortality, also zinc deficiency and both are the most leading causes…

Yes it is not the only deficiency others also have their own great contribution. But
in this paragraph we tried to be specific to vitamin A. We have modified

4. Line 113 : … vulnerable population segment by for those who have inadequate intake
of vitamin A from food

Modified accordingly

5. Line 114 : lactating women are the most… in addition to children under 5y old.

Modified accordingly

6. Line 115 : … amount of micronutrients… I think here you have to put enough of vit
A

Modified accordingly

7. Line 118… line 122 can be deleted

We believe it will give some insight

8. Line 129 : « …remains… » is there any measures taken nationally?

There was a measure taken to prevent this problem by preparing guidelines for the
prevention and control of micronutrient deficiencies at national level

9. Line 132 : … of reproductive age… what about lactating women ? is there any
regional or national statistics ?

We didn’t get data which represents lactating women at regional or national level

10. Authors can reinsert here paragraph starting by line 98 before line 134

Modified accordingly

11. Line 138 to line 144 , this paragraph needs edits and rephrasing (children who
born with low vit A reserves and from deficient mothers are at greater risk to
develop a vit A deficiency

Modified accordingly

12. Line 202 : adapted questionnaire instead of pretested questionnaire

Modified accordingly

13. Line 206 : .. and other litteratures, can the authors be more precise

Modified accordingly

14. I suggest to authors that data quality assurance comes after data analysis

Modified accordingly

15. Line 226 running frequency ????

Modified accordingly, it was not done for this research; we removed the statement
from the paragraph.

16. Line 250 was obtained instead of secured

Modified accordingly

17. Table 1 did authors collect information about household income ??

Not collected. We have collected the wealth index data

18. I think that the discussion needs a more in-depth, and authors have to present
their major findings related to the population studied and not children aged between
6 to 23 months… to compare them to other studies conducted among lactating women
(and explain in details the results obtained)

Authors may add more statements in the end to highlight the implication and
conclusions from this study

Thank you for the concerns, the major limitation for this research was shortage of
literatures. We tried to amend according to the comment.

to editor and reviewers.docx
---

## [Decision Letter · Decision Letter 2]

4 Jul 2020

PONE-D-19-28467R2

Adequate vitamin A rich food consumption and associated factors among lactating
mothers visiting child immunization and post-natal clinic at health institutions in
Gondar Town, Northwest Ethiopia

PLOS ONE

Dear Dr. Aserese,

Thank you for submitting your manuscript to PLOS ONE. After careful consideration, we
feel that it has merit but does not fully meet PLOS ONE’s publication criteria as it
currently stands. Therefore, we invite you to submit a revised version of the
manuscript that addresses the points raised during the review process.

As you will see below, one of the reviewers still requires some minor revisions to
the paper. If you are able to complete these satisfactorily we should be closer to
accepting your manuscript.

Please submit your revised manuscript by Aug 18 2020 11:59PM. If you will need more
time than this to complete your revisions, please reply to this message or contact
the journal office at plosone@plos.org. When
you're ready to submit your revision, log on to https://www.editorialmanager.com/pone/ and select the 'Submissions
Needing Revision' folder to locate your manuscript file.

If you would like to make changes to your financial disclosure, please include your
updated statement in your cover letter. Guidelines for resubmitting your figure
files are available below the reviewer comments at the end of this letter.

We look forward to receiving your revised manuscript.

Kind regards,

Robin D Clugston, Ph.D.

Academic Editor

PLOS ONE

Reviewers' comments:

Reviewer's Responses to Questions

**Comments to the Author**

1. If the authors have adequately addressed your comments raised in a previous round
of review and you feel that this manuscript is now acceptable for publication, you
may indicate that here to bypass the “Comments to the Author” section, enter your
conflict of interest statement in the “Confidential to Editor” section, and submit
your "Accept" recommendation.

Reviewer #2: All comments have been addressed

2. Is the manuscript technically sound, and do the data
support the conclusions?

Reviewer #2: Yes

3. Has the statistical analysis been performed
appropriately and rigorously? 

Reviewer #2: I Don't Know

4. Have the authors made all data underlying the
findings in their manuscript fully available?

Reviewer #2: Yes

5. Is the manuscript presented in an intelligible
fashion and written in standard English?

Reviewer #2: Yes

6. Review Comments to the Author

Reviewer #2: All questions raised in my previous review were addressed by authors.
The paper looks better after the author's correction. I have made some suggestions
to improve it more ...

Methods

I suggest to authors to add a paragraph about the HKI-FFQ as it is the main tool used
in this paper at the Sampling method part (like it has been validated by WHO using
serum retinol levels to provide reliable estimates of subclinical VAD.

A brief description of the tool and then a paragraph about how authors have adapted
the tool to local conditions ( what are questions that has been added or modified,
steps )

Discussion

needs some edits there is some suggestions :

line 282 : our findings showed that the consumption of adequate vitamin A rich foods
was….

Line 283 : lactating women should have a slightly higher vitA requirements compared
to women who are not pregnant or lactating as vitA in mother’s diet are directly
transferred into breast milk.

Line 287 : our results are higher than those found in Danot, Rural Kenya, ethiopia..
Indeed, in the Damot sore district, the study revealed that only 12,5% …..

In rural kenya, only 13,5%.....

While in India, the study revealed that …

This difference in results can be explained by the effect of sample size, study
settings , physiological state…..

Please delete the time gap btw studies…

Please delete paragraph 308…312 « no add value »

Line 314 please modify unable to write and read by illiterate mothers (in all the
manuscript)

Line 316 : educational status by educational level

Line 317 : because by may be explained by the fact that …

Line 319 : a family size <= 3 had 4 times the most likely chance to…

Line 320 : this may be due to a difficulty to provide….

Please change “mothers may give…. By « mothers may sacrifice their own
quality/quantity of nutrition in order to protect their children.

Line 324 : this may be due to that mothers with low socioeconomic status may not have
access or can afford to buy food rich in vitA…there is a strong and consistent
evidence that food insecure mothers are at higher risk of malnutrition.

Limitation

Another limitation of this study is serum retinol levels were not determined , which
could not be done ( limited budget, lack of required facilities….)

Conclusion

Line 347:

Based on the HK threshold values, 61% of lactating women have inadequate vitA food
consumption (not 38,9%) and thus, was not a public health problem among our
population.

Line 349: having college educational level and above,

7. PLOS authors have the option to publish the peer
review history of their article (what does this mean?). If published, this will
include your full peer review and any attached files.

If you choose “no”, your identity will remain anonymous but your review may still be
made public.

**Do you want your identity to be public for this peer review?** For
information about this choice, including consent withdrawal, please see our
Privacy Policy.

Reviewer #2: **Yes: **Asmaa EL HAMDOUCHI

---

## [Author Response · Author response to Decision Letter 2]

22 Jul 2020

Dear Editor and Reviewer 

Sincerest thanks for your response and reviewers comments on our manuscript. We have
modified the paper in response to the extensive and insightful editor and reviewer
comments. We would be glad to respond to any further questions and comments that you
may have.

Response to Editor and Reviewer

1. I suggest to authors to add a paragraph about the HKI-FFQ as it is the main tool
used in this paper at the Sampling method part (like it has been validated by WHO
using serum retinol levels to provide reliable estimates of subclinical VAD.

A brief description of the tool and then a paragraph about how authors have adapted
the tool to local conditions ( what are questions that has been added or modified,
steps )

The HKI FFM asks respondents how many days in the last week consumed the
foods listed on a predesigned FFQ. Only major sources of vitamin A are taken into
consideration (t100 RE), though some attention are given to other foods. The tool
has 28 different food item questions. From the 28 food items we used 21 food items
which are available in the study area and consumed by the society, the remaining 7
food items like, noodles, Amaranth leaves, sweet potato leaves, cod liver oil,
coconuts, weaning food fortified with vitamin A, Margarine fortified with vitamin A
were not included in the tool because they are not commonly consumed in the study
area.

2. Discussion needs some edits there is some suggestions :

line 282 : our findings showed that the consumption of adequate vitamin A rich foods
was….

Line 283 : lactating women should have a slightly higher vitA requirements compared
to women who are not pregnant or lactating as vitA in mother’s diet are directly
transferred into breast milk.

Line 287 : our results are higher than those found in Danot, Rural Kenya, ethiopia..
Indeed, in the Damot sore district, the study revealed that only 12,5% …..

In rural kenya, only 13,5%.....

While in India, the study revealed that …

This difference in results can be explained by the effect of sample size, study
settings , physiological state…..

Please delete the time gap btw studies…

Please delete paragraph 308…312 « no add value »

Line 314 please modify unable to write and read by illiterate mothers (in all the
manuscript)

Line 316 : educational status by educational level

Line 317 : because by may be explained by the fact that …

Line 319 : a family size <= 3 had 4 times the most likely chance to…

Line 320 : this may be due to a difficulty to provide….

Please change “mothers may give…. By « mothers may sacrifice their own
quality/quantity of nutrition in order to protect their children.

Line 324 : this may be due to that mothers with low socioeconomic status may not have
access or can afford to buy food rich in vitA…there is a strong and consistent
evidence that food insecure mothers are at higher risk of malnutrition.

Limitation

Another limitation of this study is serum retinol levels were not determined , which
could not be done ( limited budget, lack of required facilities….)

Conclusion

Line 347:

Based on the HK threshold values, 61% of lactating women have inadequate vitA food
consumption (not 38,9%) and thus, was not a public health problem among our
population.

Line 349: having college educational level and above,

All the comments and suggestion has been addressed in the document.

editor and reviewers.docx
---

## [Editor Report · Decision Letter 3]

30 Jul 2020

PONE-D-19-28467R3

Adequate vitamin A rich food consumption and associated factors among lactating
mothers visiting child immunization and post-natal clinic at health institutions in
Gondar Town, Northwest Ethiopia

PLOS ONE

Dear Dr. Aserese,

Thank you for submitting your manuscript to PLOS ONE. After careful consideration, we
feel that it has merit but does not fully meet PLOS ONE’s publication criteria as it
currently stands. Therefore, we invite you to submit a revised version of the
manuscript that addresses the points raised during the review process.

Thank you for addressing all of the previous reviewers comments and resubmitting your
revised manuscript. As originally communicated to you in my June 18th decision
letter, I have one remaining concern with the manuscript and that is the quality of
the written English. As previously indicated, the manuscript requires careful
copy-editing before it can be accepted. I have copied the text from the journal's
publication criteria below so you can better understand this policy:

*"PLOS ONE* does not copyedit accepted manuscripts, so the language in
submitted articles must be clear, correct, and unambiguous. We may reject papers
that do not meet these standards.

If the language of a paper is difficult to understand or includes many errors, we may
recommend that authors seek independent editorial help before submitting a revision.
These services can be found on the web using search terms like “scientific editing
service” or “manuscript editing service.”

Please submit your revised manuscript by Sep 13 2020 11:59PM. If you will need more
time than this to complete your revisions, please reply to this message or contact
the journal office at plosone@plos.org. When
you're ready to submit your revision, log on to https://www.editorialmanager.com/pone/ and select the 'Submissions
Needing Revision' folder to locate your manuscript file.

If you would like to make changes to your financial disclosure, please include your
updated statement in your cover letter. Guidelines for resubmitting your figure
files are available below the reviewer comments at the end of this letter.

We look forward to receiving your revised manuscript.

Kind regards,

Robin D Clugston, Ph.D.

Academic Editor

PLOS ONE

---

## [Author Response · Author response to Decision Letter 3]

7 Aug 2020

Dear Editor and Reviewer 

Sincerest thanks for your response and comments on our manuscript. We have modified
the quality of English in this paper in response to the given comments. We would be
glad to respond to any further questions and comments that you may have.

to editor and reviewers.docx
---

## [Editor Report · Decision Letter 4]

13 Aug 2020

PONE-D-19-28467R4

Adequate vitamin A rich food consumption and associated factors among lactating
mothers visiting child immunization and post-natal clinic at health institutions in
Gondar Town, Northwest Ethiopia

PLOS ONE

Dear Dr. Aserese,

Thank you for submitting your manuscript to PLOS ONE. After careful consideration, we
feel that it has merit but does not fully meet PLOS ONE’s publication criteria as it
currently stands. Therefore, we invite you to submit a revised version of the
manuscript that addresses the points raised during the review process.

As previously communicated, I am still not satisfied with the quality of the
manuscript's written English and have made the following specific suggestions to
help improve the manuscript.

Line 24: …low-income countries and is a major…

Line 30: An institution-based…

Line 31: …March 2019, and included 631 study participants. (delete “were included in
the study”)

Line 74: “Vitamin A deficiency is a major public health problem, especially among
low-income countries”

Line 77: 5 years (insert space, add ‘s’)

Line 79: …to replace loss that occurs through…

Line 80: Please insert a space between the value and the unit

Line 81: insert comma after the word foods

Line 81: replace “which” with “that”

Line 83: insert comma after the word products

Line 86: Little attention has been given to the problem of vitamin A deficiency in
breastfeeding as compared to preschool children because the clinical symptoms of
xerophthalmia in women are rare.

Line 91: …deficiency has been demonstrated in different African countries. For
example, about 40%...

Line 94: In Ethiopia, research has shown that vitamin A deficiency remains a public
health concern, including a high prevalence of Bitot’s spots, with the highest…

Line 99: replace “better” with “optimal”

Line 100: Several causes of vitamin A deficiency in pregnant women have been
identified, including inadequate consumption of vitamin A-rich foods,
socio-demographic factors, economic status, duration of lactation (prolonged),
amount of body fat, and hemoglobin concentration (2, 11, 12, 14, 20).

Line 103: No need to start new paragraph here

Line 104: Vitamin A deficiency has a wide range of consequences, with the most common
manifestations including ocular lesions (e.g. xerophthalmia), mortality, failure to
thrive, and poor reproduction (21).

Line 107: Children born with…

Line 109: No need to start new paragraph here. Change text: “Attempts to prevent and
control vitamin A deficiency include promoting

Line 110: remove comma between “mothers” and “and”

Line 111: remove comma between “conditions” and “and”

Line 111: “fortification of cooking oil with vitamin A”

Line 113: …”launched a revised”. It would be helpful to
include the data here. For example “In 20XX, Ethiopia launched a revised…”

Line 117: “objective, which has made a great contribution to preventing vitamin A
deficiency”

Line 125: “750 km”

Line 126: …health office report, it is estimated that there are currently…

Line 136: No need to start new paragraph here

Line 140. No need to start new paragraph here

Line 143-146: Here you introduce the abbreviation HC for health center (line 143) but
do not use it in line 145 and 146. Please be consistent with the use of this
abbreviation, or do not introduce it.

Line 149: Each of these definitions do not need to appear on a new line and can be
consolidated into a single paragraph. Also, I do not recommend you using bold for
the operational terms; maybe use italics so it does not conflict with the journal’s
formatting guidelines. I recommend starting this paragraph like this: “The following
operational definitions were used in this study. *Adequate…”*

Line 163: Insert FFQ as abbreviation for food frequency questionnaire here

Line 170: Please define what FFM stands for here. Spell it out.

Line 170: “…last week they consumed…”

Line 182: No need to start new paragraph here

Line 188: “A principal component…”

Line 191: delete comma

Line 218: delete “have”

Line 219: insert space after 27

Line 252: do not capitalize “meat”

Line 263: replace “to” with “for”

Line 268: “…were included in the multivariable analysis. Educational…”

Line 270: “...considered to be determinants for…”

Line 276: “…status was found…”

Line 277: “vitamin A-rich food…” Start sentence with capital letter: “Mothers…”

Line 290: Please check for meaning > “Lactating women have a higher vitamin A
requirement compared to women who are not pregnant as vitamin A in the mother’s diet
is directly transferred into breast milk. As such, the FAO/WHO…”

Line 300: “…, and physiological state”

Line 301: No need to start a new paragraph here.

Line 304: “This difference may reflect differing economic status, as the study
participants…”

Line 312: “…consequences of inadequate dietary vitamin A intake”

Line 313: “..4 times more likely chance of consuming adequate…”

Line 315: “…foods to a large family…”

Line 318: “This may be because mothers…”

Line 322: “WHO guidelines”

Line 325: No need to start a new paragraph here

Line 332: “Because the study was institution based, it may overestimate consumption
and may…

Throughout the entire manuscript, please make sure there is a space between the last
letter and the brackets containing the references. For example, line 75 has no
space, but line 76 does have a space. All brackets containing references should be
preceded by a single space.

Please submit your revised manuscript by Sep 27 2020 11:59PM. If you will need more
time than this to complete your revisions, please reply to this message or contact
the journal office at plosone@plos.org. When
you're ready to submit your revision, log on to https://www.editorialmanager.com/pone/ and select the 'Submissions
Needing Revision' folder to locate your manuscript file.

If you would like to make changes to your financial disclosure, please include your
updated statement in your cover letter. Guidelines for resubmitting your figure
files are available below the reviewer comments at the end of this letter.

We look forward to receiving your revised manuscript.

Kind regards,

Robin D Clugston, Ph.D.

Academic Editor

PLOS ONE

---

## [Author Response · Author response to Decision Letter 4]

1 Sep 2020

Response to Reviewers

Dear respected editor,

Sincerest thanks for your response and comments on our manuscript. We have modified
the quality of English in this paper in response to the given comments. We would be
glad to respond to any further questions and comments that you may have. We look
forward to hear your response. Thank you!

to Reviewers.docx
---

## [Editor Report · Decision Letter 5]

4 Sep 2020

Adequate vitamin A rich food consumption and associated factors among lactating
mothers visiting child immunization and post-natal clinic at health institutions in
Gondar Town, Northwest Ethiopia

PONE-D-19-28467R5

Dear Dr. Aserese,

We’re pleased to inform you that your manuscript has been judged scientifically
suitable for publication and will be formally accepted for publication once it meets
all outstanding technical requirements.

Kind regards,

Robin D Clugston, Ph.D.

Academic Editor

PLOS ONE
---

## [Editor Report · Acceptance letter]

11 Sep 2020

PONE-D-19-28467R5

Adequate vitamin A rich food consumption and associated factors among lactating
mothers visiting child immunization and post-natal clinic at health institutions in
Gondar Town, Northwest Ethiopia

Dear Dr. Aserese:

I'm pleased to inform you that your manuscript has been deemed suitable for
publication in PLOS ONE. Congratulations! Your manuscript is now with our production
department.

Kind regards,

on behalf of

Dr. Robin D Clugston 

Academic Editor

PLOS ONE